# Long-Term Outcomes of Endoscopic Submucosal Dissection for Colorectal Epithelial Neoplasms: A Systematic Review

**DOI:** 10.3390/cancers15010239

**Published:** 2022-12-30

**Authors:** Toshihiro Nishizawa, Takashi Ueda, Hirotoshi Ebinuma, Osamu Toyoshima, Hidekazu Suzuki

**Affiliations:** 1Department of Gastroenterology and Hepatology, International University of Health and Welfare, Narita Hospital, Narita 286-8520, Japan; 2Gastroenterology, Toyoshima Endoscopy Clinic, Tokyo 157-0066, Japan; 3Division of Gastroenterology and Hepatology, Department of Internal Medicine, Tokai University School of Medicine, Isehara 259-1193, Japan

**Keywords:** colorectal, endoscopic submucosal dissection, long-term outcomes

## Abstract

**Simple Summary:**

Endoscopic submucosal dissection (ESD) facilitates a successful en bloc resection regardless of tumor size. In this review, we summarize up-to-date reports with long-term observation after ESD for colorectal epithelial neoplasms. The strategy of ESD and additional surgery depending on the curative status showed an excellent five-year disease-specific survival rate. Incomplete resection is a risk factor for local recurrence, and endoscopists must improve their skill level. In non-curative ESD, optimization of additional surgery could reduce disease-specific death without additional surgery.

**Abstract:**

In this review, we summarize up-to-date reports with five-year observation after colorectal endoscopic submucosal dissection (ESD). Five-year cause-specific survival rates ranged from 98.6 to 100%. The local recurrence rates ranged from 1.1 to 2.2% in complete resection and 7.5 to 25.0% in incomplete resection. Incomplete resection was a risk factor for local recurrence. In non-curative ESD, five-year cause-specific survival rates ranged from 93.8 to 100% with additional surgery, and 92.7 to 99.1% without surgery. The choice of additional surgery should be based on the individual patient’s age, concomitant diseases, wishes, life expectancy, and the risk of lymph node metastasis. The metachronous cancer rates ranged from 0.22 to 1.1%. Both local recurrence and metachronous tumors should be checked with a follow-up colonoscopy after ESD.

## 1. Introduction

Colorectal cancer is a leading cause of cancer-related death [1]. By 2030, approximately two million new colorectal cancer cases and one million related deaths are predicted to occur [2]. However, its mortality and morbidity can be reduced by colonoscopy and subsequent removal of its precursor lesions [3]. Endoscopic mucosal resection (EMR) and endoscopic submucosal dissection (ESD) are the most common approaches for endoscopic removal of colorectal neoplasms. Even though EMR is safe and convenient, lesions > 20 mm are often removed inadequately, which results in a high risk of local recurrence. On the other hand, ESD facilitates a successful en bloc resection regardless of tumor size [4,5,6].

A recent meta-analysis compared the outcomes of EMR and ESD for colorectal lesions >20 mm [7]. The en bloc resection rates in EMR and in ESD were 34.9% and 89.9%, respectively, and the complete resection rates were 36.2% and 79.6%, respectively. There were significant differences between the EMR and ESD groups (*p* < 0.001). However, the perforation rate in ESD was significantly higher than that in EMR (4.9% versus 0.9%).

ESD was developed in Japan in the late 1990s, and gastric ESD received Japanese health insurance approval in 2006 [8]. Unlike gastric ESD, a thinner colonic wall increases the risk of perforation, and colonic ESD is more difficult due to the maneuverability [4]. After the refinement of ESD techniques and devices [9,10], colorectal ESD finally got Japanese health insurance approval in 2012.

The safety and efficacy of ESD as a minimally invasive treatment for gastrointestinal neoplasms has been well-established, especially in Japan and other Asian countries. Although colorectal ESD is technically demanding, it can safely be performed by experts using the appropriate devices and techniques. Figure 1 shows colorectal ESD using pocket creation method. This technique allows stable scope position and sufficient tissue traction inside the pocket. Figure 2 shows clip-with-loops method. The feature of this method is that counter traction is generated using loops, one side of which is attached to the edge of the lesion with a clip, and the other side is attached to the opposite side of the colonic wall. Multi-loops allow to change the direction of traction by adding clip or cutting off the loop [11].

Recently, several studies have evaluated the long-term outcomes of colorectal ESD with five-year observation. This review summarizes up-to-date reports about the long-term outcomes of colorectal ESD.

## 2. Literature Search and Selection

The literature was systematically searched using PubMed and Cochrane Library databases, Web of Science, and the Igaku–Chuo–Zasshi database in Japan (up to December 2022). The search words were: “colorectal or colonic or large intestine” AND “endoscopic submucosal dissection or ESD” AND “long-term or long period or five-year or 5-year”. 

The inclusion criteria were as follows: (1) study design: cohort study, (2) participants: patients who had colorectal epithelial neoplasms, (3) intervention: ESD, and (4) outcome: survival or recurrence rate in long-term observation. The exclusion criteria were as follows: (1) median follow-up period of <36 months, (2) outcomes including EMR, (3) studies limited to inflammatory bowel disease-related neoplasms, residual lesions, or rectal lesions, and ESD with additional surgery, (4) review articles, (5) meeting abstracts, (6) case reports, and (7) duplication. 

Six hundred and twenty-six citations were retrieved by the literature search process. Among these, we excluded 537 studies based on the exclusion criteria (239 unrelated topics, 112 review articles, 75 meeting abstracts, 35 case reports, 61 duplications, and 15 protocols from prospective studies). The remaining 89 studies were reviewed and 75 studies were excluded (34 studies with a median follow-up period of <36 months, 16 involving EMR, eight limited to inflammatory bowel disease-related neoplasms, seven limited to ESD with additional surgery, eight limited to rectal lesions, and two limited to residual lesions). Finally, we included 14 studies in this review (Figure 3) [12,13,14,15,16,17,18,19,20,21,22,23,24,25].

Statistical analysis was performed using Comprehensive Meta-Analysis (CMA) software (version 4, Biostat, Inc., Englewood, NJ, USA). Pooled event rates and 95% confidence intervals (CIs) were calculated using a random-effects model [26].

## 3. Five-Year Overall Survival and Five-Year Disease-Specific Survival after Colorectal ESD

There are five reports describing overall and disease-specific survival after colorectal ESD at five years (Table 1).

Cong et al. in China assessed the long-term outcome of ESD for colorectal laterally spreading tumors larger than 30 mm [12]. A needle-knife, a hook-knife, and/or an insulated-tip knife (Olympus, Tokyo, Japan) were used. The injection solution was a mixture of saline, indigo carmine and epinephrine. The mean lesion size was 52 mm. The en bloc resection rate was 83.1% (147 of 177 lesions), and the complete resection rate was 81.4% (144 of 177). Perforation rate was 2.3% (4 cases). Three intraoperative perforations were treated with endoscopic clipping and antibiotics. One delayed perforation was treated by a long period of fasting and antibiotics. Delayed bleeding was observed in six cases (3.4 %). One case was stopped after conservative treatment, and the other cases were controlled by endoscopic hemostasis. Histologically, 13 tumors (7.3%) were submucosal invasive cancer. When histology confirmed submucosal invasion, surgical resection was recommended. Surgery was added in nine of the 13 required cases (including three cases with positive vertical margin and two cases with piecemeal resection). Local recurrence occurred in 11 cases (7.7%), and they were cured by ESD or surgery. None of the patients died of colorectal cancer. The five-year overall survival rate was 95.5%, and the five-year disease-specific survival rate was 100%. 

The other four reports were from Japan. According to the Japanese Gastroenterological Endoscopy Society guidelines, curative resection is defined as follows: (i) complete histological resection; (ii) papillary or tubular adenocarcinoma; (iii) submucosal invasion depth <1000 μm; (iv) no vascular invasion; and (v) tumor budding grade 1 [27,28].

Kuwai et al. assessed the long-term outcomes of colorectal ESD using Stag-beetle Knife Jr [29]. The injection solution was sodium hyaluronate (Johnson & Johnson, New Jersey, USA). The mean lesion size was 34.3 mm. The en bloc resection rate was 98.4% (243 of 247 lesions). The curative resection rate was 85.4% (211 of 247 lesions). Perforation rate was 0.4% (one case). The case with delayed perforation was treated conservatively. Delayed bleeding was observed in six cases (2.4%) and was controlled by endoscopic hemostasis. Histologically, 26 tumors (11%) were deep submucosal invasive cancer. The local recurrence rate was 1.1% (two of 187 lesions in the follow-up cohort). One patient died of colorectal cancer. In this patient, histology confirmed complete resection, but noncurative resection. Despite additional surgery, lymph node and distant metastasis occurred one year after surgery. The five-year overall survival rate was 94.1%, and the five-year disease-specific survival rate was 98.6%. 

Takahashi et al. compared the long-term outcomes of colorectal ESD in non-elderly (<75 years of age) and elderly (≥75 years of age) patients [14]. A dual knife and hook knife (Olympus, Tokyo, Japan), and/or flush knife (Fujinon Co., Tokyo, Japan) were used. The injection solution was 10% glycerol and/or sodium hyaluronate. The curative resection rates were 93.2% in the non-elderly group and 92.6% in the elderly group. Perforation rates were 5.9% in the non-elderly group and 6.7% in the elderly group. Emergency surgery was required due to intraoperative perforation (one case; 0.3%) and intraoperative bleeding (one case; 0.3%) in the non-elderly group. In the elderly group, emergency surgery was required due to delayed perforation (one case; 0.6%) and delayed bleeding (one case; 0.6%). Histologically, 19 tumors (6%) were deep submucosal invasive cancer in the non-elderly group. In the elderly group, 11 tumors (7%) were deep submucosal invasive cancer. Surgery was added for 15 of 23 required cases (65.2%) in the non-elderly group and for seven of the required 12 (58.3%) in the elderly group. Local recurrence rates were 0% in the non-elderly group and 2.0% in the elderly group. None of the patients died of colorectal cancer. The five-year disease-specific survival rates were 100% in both groups. The five-year overall survival rate in the elderly group (86.3%) was significantly lower than that in the non-elderly group (93.5%; *p* < 0.05). 

Boda et al. conducted a large multicenter study [15]. ESD was performed using 1 or 2 ESD knives including a dual knife, hook knife, IT knife and flex knife (Olympus, Tokyo, Japan), flush knife, and SB knife Jr. The mean lesion size was 33 mm. The curative resection rate was 83.7% (1054 of 1259 lesions). Perforation rate was 3.8%. Ten patients (0.8%) required emergency surgery owing to perforation. Delayed bleeding after ESD occurred in 46 patients (3.7%). Histologically, 153 tumors (12.1%) were deep submucosal invasive cancer. The curative resection rate was 83.7% (1054 of 1259 lesions). Surgery was added for 134 of 205 required cases (65.4%). The local recurrence rate was 1.7% (14 of 840 lesions in the follow-up cohort). Two patients died of colorectal cancer. In a 46-year-old female, histology confirmed complete resection but deep submucosal invasion (1900 µm). Without additional surgery, the patient died 71 months after ESD due to liver metastasis. In a 75-year-old male with rectal cancer, histology confirmed complete resection but deep submucosal invasion (3500 µm) and vascular invasion. Without additional surgery, the patient died 19 months after ESD due to lung metastasis. The five-year overall and disease-specific survival rates were 92.3% and 99.6%, respectively. 

Ohata et al. conducted a prospective multicenter study [16]. ESD was performed using 1 or 2 ESD knives including a dual knife, IT knife nano (Olympus, Tokyo, Japan), flush knife, bipolar needle knife (Xeon Medical, Tokyo, Japan), and Mucosectom (Hoya. Tokyo, Japan). The injection solution was mainly sodium hyaluronate. The mean lesion size was 32.4 mm. The en bloc resection rate was 97.0% (1759 of 1814 lesions). The curative resection rate was 78.9% (1431 of 1814 lesions). Seven patients (0.4%) required emergency surgery owing to delayed perforation. Histologically, 145 tumors (8.0%) were deep submucosal or deeper invasive carcinomas. Surgery was added for 111 of 157 required cases (72.1%). The local recurrence rate was 0.5% (eight of 1640 lesions in the follow-up cohort). Four patients died of rectal cancer. In three patients, histology confirmed positive vertical margin and vascular invasion. Despite additional surgery, distant metastasis occurred between 14 and 25 months after ESD. In one patient (an 83-year-old male), histology confirmed positive horizontal margin and deep submucosal invasion (6000 µm). Without additional surgery, distant metastasis occurred 13 months after ESD. The five-year overall disease-specific survival rates were 93.6% and 99.6%, respectively. 

Pooled event rates (95% CIs) were 94.6% (90.5–97%) for en bloc resection, 87.1% (82.1–90.8%) for curative resection, 2.4% (1.1–5.1%) for perforation, 1.6% (0.6–4.4%) for local recurrence, 93.0% (89.6–95.4%) for five-year overall survival, and 99.4% (99.1–99.7%) for five-year overall disease-specific survival.

To summarize, the strategy of ESD and additional surgery depending on the curative status showed an excellent five-year disease-specific survival rate.

## 4. Five-Year Local Recurrence after Colorectal ESD

There are four reports describing five-year local recurrence after colorectal ESD (Table 2).

In Japan, Yamada et al. analyzed the long-term outcome of colorectal ESD [17]. The mean lesion size was 37 mm. The complete resection rate was 81% (344 of 423 lesions). Perforation rate was 3%. Histologically, 104 tumors (25%) were low-grade adenomas, 225 (53%) were high-grade dysplasia, 41 (10%) were superficial submucosal invasive cancer, and 53 (13%) were deep submucosal invasive cancer. The rate of submucosal cancer was 23%. The five-year local recurrence was 3.8%. Their multivariate analysis showed that piecemeal resection and deep submucosal cancer were major risk factors for local recurrence.

Arribas et al. in Spain reported the long-term outcome of ESD in non-Asian countries [18]. The median size was 33 mm. The en bloc resection rate was 60.9%, and the complete resection rate was 30.4%. The perforation rate was high at 11.6%. Histologically, 23 tumors (33.4%) were low-grade adenomas, 45 (65.2%) were high-grade dysplasia, and 1 (1.4%) was superficial submucosal invasive cancer. The five-year local recurrence was also high, at 14.5%.

A recent meta-analysis showed that the outcomes of colorectal ESD in Asian countries were significantly better than those in non-Asian countries: 85.6% versus 71.3% in complete resection rate, 93% versus 81.2% in en bloc resection rate, 4.5% versus 8.6% in perforation rate, and 0.8% versus 3.1% in emergent surgery due to ESD-related adverse events [30]. This meta-analysis concluded that the ESD procedure in non-Asian countries has not yet reached adequate performance levels. In Europe, major obstacles for the dissemination of colorectal ESD include few suitable starting cases in the stomach, few experts, long learning curve, high risk of complications, and lack of standardized step-up training programs [31]. Recently, better en bloc rates of 87.9–89.5% have been reported from several groups in Europe, especially high-volume centers [32,33]. 

Park et al. in Korea also analyzed the long-term outcome of colorectal ESD [19]. The mean lesion size was 28 mm. The en bloc resection rate and complete resection rate were 88.4% and 80.6%, respectively. Perforation rate was 4.8%. Histologically, 436 tumors (56.1%) were adenomas, 18 (2.3%) were serrated lesions, 239 (30.7%) were mucosal cancer, and 85 (10.9%) were submucosal cancer. The five-year local recurrence was 2.2%. 

Qu et al. in China assessed the long-term outcomes of ESD for early colorectal cancer [20]. The mean lesion size was 24 mm. The en bloc resection was 89.2%. Perforation rate was 1.5%. Histologically, 15 tumors (23.1%) were mucosal cancer, and 50 tumors (76.9%) were submucosal cancer. The five-year local recurrence rates were 7.7%. The high submucosal cancer rate (76.9%) may have influenced the high local recurrence rate.

These data indicated that incomplete resection and deep submucosal cancer were risk factors for local recurrence.

## 5. Resection Status and Recurrence Rates

Three studies reported resection status and recurrence rates with a median follow-up period of ≥36 months (Table 3) [19,21,22]. The local recurrence rates ranged from 0.4 to 3.7% in en bloc resection and 6.3 to 26.1% in non-en bloc resection. There were significant differences between en bloc and non-en bloc resection in each study. The local recurrence rates ranged from 1.1 to 2.2% in complete resection and 7.5 to 25.0% in incomplete resection [19,22]. There were significant differences between complete resection and incomplete resection in each study. 

These data reconfirmed that incomplete resection was a risk factor for local recurrence.

## 6. Non-Curative Resection with and without Surgery, and Long-Term Outcomes

Two studies compared additional surgery and surveillance without surgery in non-curative ESD (Table 4).

Kato et al. analyzed the long-term outcomes of non-curative colorectal ESD in elderly patients (≥75 years) [23]. For non-curative resection, additional surgery was recommended. However, the background and wishes of patients were also considered. The mean age was 78 years in the additional surgery group and 81 years in the surveillance without surgery group. Reasons for non-curative ESD were shown in Table 4. Despite no recurrence in the additional surgery groups, surveillance without surgery groups had local recurrence in 4.6% and metastatic recurrence in 5.7%. The five-year overall and disease-specific survival rates were 86.6% and 100%, respectively, in the additional surgery group, and 76.6% and 96.3%, respectively, in the surveillance without surgery group. There was a significant decrease in disease-specific survival rates in the surveillance without surgery group (*p* = 0.006). These data suggested that even in elderly patients, avoidance of additional surgery increased the risk of colorectal cancer metastasis and subsequent death. Their multivariate analysis showed that low (<96.3) geriatric nutritional risk index was the sole risk factor for reduced overall survival. Positive lymphatic involvement was most significantly associated with the risk of metastasis. The authors suggested that geriatric nutritional risk index and lymphatic involvement should be taken into account when deciding the therapeutic strategy after non-curative colorectal ESD in order to balance the risk of colorectal cancer-related and non- colorectal cancer-related mortality.

Li et al. analyzed the long-term outcome of non-curative ESD [24]. For non-curative resection, additional surgery was recommended. However, some patients chose surveillance without surgery because of old age or preservation of the anus. The additional surgery group was significantly younger than the surveillance without surgery group. Reasons for non-curative ESD were shown in Table 4. Local and metastatic recurrence in the surveillance without surgery group tended to be higher than those in the additional surgery group. The five-year disease-specific survival rates were 93.8% and 92.7% in the additional surgery and surveillance without surgery groups, respectively. There were no significant differences between these two groups. The authors suggested that surveillance without surgery after non-curative ESD may serve as good alternatives to additional surgery, especially in patients with more advanced ages.

There is an interesting study, although it included EMR and ESD [34]. Tamaru et al. also reported the comparison between with and without surgery in non-curative ESD [34]. Surveillance without surgery group was significantly older than the additional surgery group (69.3 years versus 63.3 years). The five-year overall survival in the surveillance without surgery group was significantly lower than that in the additional surgery group (79.3% versus 92.4%). On the other hand, there were no differences between with and without surgery in five-year disease-specific survival rates (98.3% versus 99.1%) and recurrence rates (2.5% versus 3.3%). The authors emphasized the importance of deciding on additional surgery, and the considerations were the individual patient’s age, concomitant diseases, wishes, life expectancy, and concrete risk of lymph node metastasis [34,35,36]. Optimization of additional surgery could reduce disease-specific death without additional surgery.

Several studies have investigated the predictive risk of lymph node metastasis [28]. The risk of lymph node metastasis in cases with only submucosal invasion ≥1000 μm and no other risk factors was reported to be 1.3% (95% confidence interval [CI], 0–2.4) [37]. A meta-analysis analyzed risk factors for lymph node metastasis and showed that the significant risks were submucosal invasion ≥1000 µm (odds ratio, 3.00; 95% CI, 1.36–6.62), lymphatic invasion (odds ratio, 6.91; 95% CI, 5.40–8.85), vascular invasion (odds ratio, 2.70; 95% CI, 1.95–3.74), and poorly differentiated carcinoma (odds ratio, 8.27; 95% CI, 4.67–14.66) [38]. A recent systematic review also stated that submucosal invasion ≥1000 µm was a weaker predictor than other risk factors [39]. Nishimura et al. analyzed 370 consecutive patients with submucosal colorectal cancer that was treated with ESD, and univariate analysis identified lymphatic invasion, poorly differentiated carcinoma, and high-grade budding as significant risk factors for lymph node metastasis [40]. Furthermore, multivariate analysis of risk factors for lymph node metastasis only identified positive lymphatic invasion (odds ratio, 3.91; 95% CI, 1.04–14.6). 

The risk of additional surgery should be balanced against the risk of lymph node metastasis, considering each patient’s life expectancy and wishes.

## 7. Metachronous Tumor after Colorectal ESD

The five-year cumulative metachronous cancer rates were reported to be 13–15% in gastric ESD [41,42] and 25–26% in esophageal ESD [13,43,44].

Three studies reporting the metachronous cancer rates after colorectal ESD were available (Table 5). Boda et al. reported 1.1% in the incidence of metachronous tumors (high grade dysplasia and cancer) [15], Ohata et al. reported 1.0% in the metachronous invasive cancer rate [16], and Shin et al. reported 0.22% in the metachronous cancer rate [25]. In the studies by Boda et al. and Ohata et al., follow-up colonoscopy was recommended once every one to two years. In Shin et al.’s study, follow-up colonoscopy was performed at least once, although a computed tomography scan was recommended once every one to two years. Therefore, Shin et al. might underestimate the metachronous cancer rate.

Although colorectal metachronous cancer is less common than gastric and esophageal ones, surveillance colonoscopy should be performed for both local recurrence and metachronous tumors after ESD.

According to the European Society of Gastrointestinal Endoscopy Guideline-Update 2022, surveillance colonoscopy is recommended at 12 months, followed by curative ESD for colorectal cancer [45]. In cases with piecemeal resection or positive lateral margins, surveillance colonoscopy is recommended at 3–6 months using high-resolution chromoendoscopy with biopsies.

## 8. Conclusions

Colorectal ESD has shown excellent five-year disease-specific survival. Incomplete resection in ESD is a risk factor for local recurrence. Endoscopists must improve their skills. Furthermore, the development of training systems will promote world-wide standardization of ESD [46,47]. In non-curative resection, the patient’s age, concomitant diseases, wishes, life expectancy, and the risk of lymph node metastasis should all be taken into account when deciding the need for additional surgery.

## Figures and Tables

**Figure 1 cancers-15-00239-f001:**
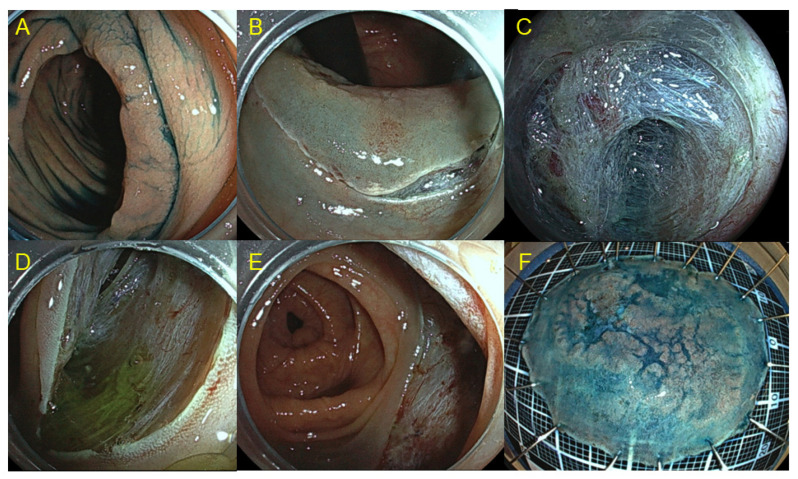
Colorectal endoscopic submucosal dissection (ESD) using pocket creation method. (**A**) Laterally spreading tumor in the ascending colon. (**B**) Mucosal incision was made around the oral side with retroflex view. (**C**) A submucosal pocket was created by dissecting the submucosa. (**D**) Submucosal tunnel was created from the anal to oral side. (**E**) Post-operative ulcer after ESD. (**F**) Specimen after ESD (complete resection, superficial submucosal invasive cancer).

**Figure 2 cancers-15-00239-f002:**
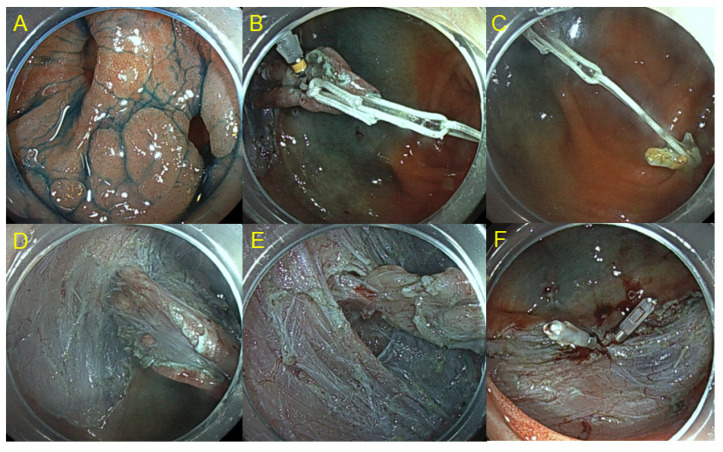
Colorectal ESD using clip-with loops method. (**A**) Laterally spreading tumor with a diverticulum in the cecum. (**B**) Clip with 2 loops was anchored to the lesion. (**C**) Another clip was attached to the opposite side of the lumen while hooking the loop. (**D**) Insufflation of CO2 enhances counter traction. (**E**) Submucosal dissection in the diverticulum. (**F**) En bloc resection was performed without any complication.

**Figure 3 cancers-15-00239-f003:**
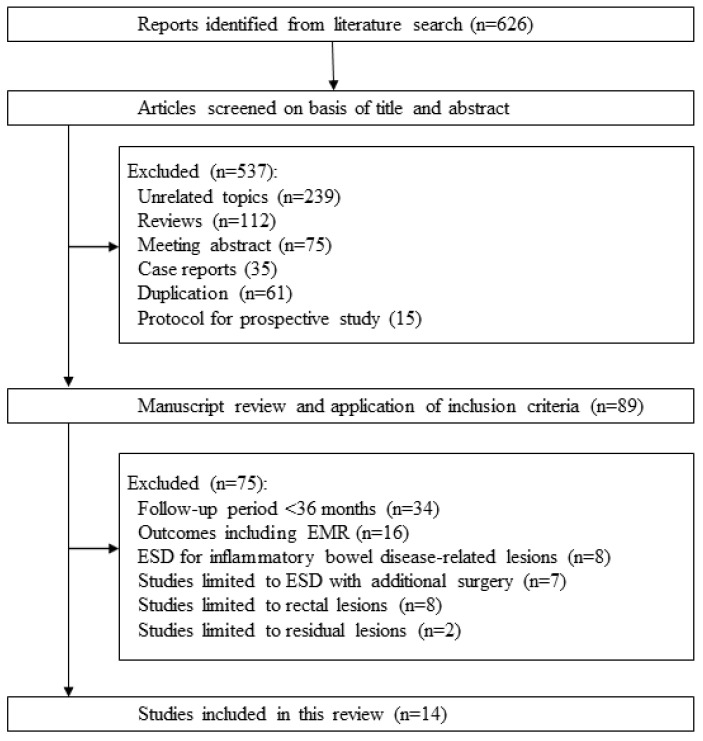
Flow diagram of the systematic literature search.

**Table 1 cancers-15-00239-t001:** Five-year overall survival and 5-year disease-specific survival after colorectal endoscopic submucosal dissection.

Author Year	Mean Age	Patient Number	En Bloc Resection	Curative Resection	Perforation Rate	Histology (Rate)	Additional Surgery Received/Required	Local Recurrence	5-Year Overall Survival	5-Year Disease Specific Survival
Cong 2016	62.9	156	83.1%	—	2.3%	Adenoma (70%)	9/13 (69.2%)	7.7%	95.5%	100%
						Serrated lesion (4%)				
						Mucosal cancer (18.6%)				
						Submucosal cancer (7.3%)				
Kuwai 2017	69.3	228	98.4%	85.4%	0.4%	Adenoma (36%)	22/36 (61.1%)	1.1%	94.1%	98.6%
						Mucosal cancer (43%)				
						Superficial SM cancer (10%)				
						Deep SM cancer (11%)				
Takahashi 2017	63.9 ‡	325	93.4%	93.2%	5.9%	Adenoma (39%)	15/23 (65.2%)	0%	93.5%	100%
						Mucosal cancer (49%)				
						Superficial SM cancer (6%)				
						Deep SM cancer (6%)				
	79.3 †	157	96.3%	92.6%	6.7%	Adenoma (32%)	7/12 (58.3%)	2.0%	86.3%	100%
						Mucosal cancer (54%)				
						Superficial SM cancer (7%)				
						Deep SM cancer (7%)				
Boda 2018	69	1233	92.6%	83.7%	3.8%	Adenoma (27%)	134/205 (65.4%)	1.7%	92.3%	99.6%
						Mucosal cancer (53%)				
						Superficial SM cancer (8%)				
						Deep SM cancer (12%)				
Ohata 2022	67.4	1814	97.6%	78.9%	0.4% *	Adenoma (27%)	111/154 (72.1%)	0.5%	93.6%	99.6%
						Mucosal cancer (53%)				
						Superficial SM cancer (8%)				
						Deep SM cancer (12%) *#*				
Pooled rate	—	—	94.6%	87.1%	2.4%	—	—	1.6%	93.0%	99.4%
95% CI			90.5–97	82.1–90.8	1.1–5.1			0.6–4.4	89.6–95.4	99.1–99.7

‡ Non-elderly group; † Elderly group: * perforation with emergent surgery; *#* Deep SM cancer or deeper; SM: submucosal; CI: confidence interval.

**Table 2 cancers-15-00239-t002:** Five-year local recurrence after colorectal endoscopic submucosal dissection.

Author Year	Country	Patients Number	En Bloc Resection	Complete Resection	Perforation Rate	Submucosal Cancer Rate	5-Year Local Recurrence
Yamada 2017	Japan	408	—	81.3%	3%	23%	3.8%
Arribas 2020	Spain	69	60.9%	30.4%	11.6%	1.4%	14.5%
Park 2021	Korea	770	88.4%	80.6%	4.8%	10.9%	2.2%
Qu 2021	China	65	89.2%	—	1.5%	76.9%	7.7%

**Table 3 cancers-15-00239-t003:** Resection status and recurrence rates.

Author Year	Patients Number	Follow-Up Period (M)	En Bloc Resection	Local Recurrence	Complete Resection	Local Recurrence
Chen 2018	610	58	En bloc	0.4% **	Complete	—
			Non-en bloc	6.9%	Incomplete	—
Park 2021	770	60	En bloc	1.7% *	Complete	1.1% ***
			Non-en bloc	6.3%	Incomplete	7.5%
Maselli 2022	327	36	En bloc	3.7% ***	Complete	2.2% ***
			Non-en bloc	26.1%	Incomplete	25.6%

M: months, *: *p* < 0.05, **: *p* < 0.01, ***: *p* < 0.001.

**Table 4 cancers-15-00239-t004:** Non-curative resection with and without surgery, and long-term outcomes.

Author Year	Country	Additional Surgery	Patient Number	Mean Age	Non-Curative Reason	5-Year Overall Survival	5-Year Disease-Specific Survival	Local Recurrence	Metastatic Recurrence
Kato 2022	Japan	With surgery	60	78	Incomplete resection (35%)	86.6%	100.0%	0%	0%
					Deep SM invasion (67%)				
					Vascular invasion (17%)				
					Lymphatic invasion (27%)				
					Undifferentiated cancer (3%)				
		Without surgery	87	81	Incomplete resection (43%)	76.6%	96.3%	4.6%	5.7%
					Deep SM invasion (61%)				
					Vascular invasion (8%)				
					Lymphatic invasion (16%)				
					Undifferentiated cancer (1%)				
Li 2021	China	With surgery	85	58	Incomplete resection (52%)	92.6%	93.8%	1.2%	3.5%
					Deep SM invasion (87%)				
					Vascular invasion (22%)				
					Undifferentiated cancer (26%)				
		Without surgery	95	61.5	Incomplete resection (54%)	92.7%	92.7%	3.2%	6.3%
					Deep SM invasion (67%)				
					Vascular invasion (15%)				
					Undifferentiated cancer (11%)				

SM: submucosal.

**Table 5 cancers-15-00239-t005:** Metachronous tumor after colorectal endoscopic submucosal dissection.

Author Year	Patients Number	Metachronous Tumor	Metachronous Tumor
Boda 2018	882	High grade dysplasia and cancer	1.1%
Ohata 2022	1437	Invasive cancer	1.0%
Shin 2022	450	Cancer	0.22%

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
