# Peer review of "Long-Term Outcomes of Endoscopic Submucosal Dissection for Colorectal Epithelial Neoplasms: A Systematic Review"

_cancers, 2022, doi:10.3390/cancers15010239_

Round 1

Reviewer 1 Report

Nishizawa et al submitted a comprehensive review on long-term outcomes of endoscopic submucosal dissection for colonic lesions. Although the article is interesting, I have some concerns on the lack of novelty as other similar papers have been already published. The authors should claim in the discussion the source of novelty of their review. 

Other comments:

 -          The article is a systematic review, so this should be specified in the title and abstract. So the title would read better as : “Long-term outcomes of endoscopic submucosal dissection for colorectal epithelial neoplasms: a systematic review”

 -          Literature search: other databases should be explored to find eventual further articles, for example Ovid, Embase, Google scholar….

 -          Please provide a reproducible search strategy with a comprehensive search string. Also, the number of key words seems to be quite low for a meaningful search

-          Why only cohort studies were included? There were no RCTs nor other prospective studies?

-          Explain how were treated perforations in the retrieved articles. Also, try to detail the outcomes of those patients who experienced major complications

-          Add some technical details of these procedures, for example submucosal injection solution, knife used and so on….

-          In Table 1, provide a pooled overall rate of the outcomes reported

-          The authors should comment deeply also on the clinical outcomes of non-curative ESD, detailing the recent evidence on that topic

-          The authors should add also a paragraph on non-Asian experience with this technique

 -          Figures are lacking, maybe some figures depicting the technical aspects of these procedures would be useful

Author Response

Reviewer 1

The article is a systematic review, so this should be specified in the title and abstract. So the title would read better as: “Long-term outcomes of endoscopic submucosal dissection for colorectal epithelial neoplasms: a systematic review”

Thank you for your important comments, which were extremely helpful for improving the quality of our manuscript. According to your comment, the title was modified to “Long-term outcomes of endoscopic submucosal dissection for colorectal epithelial neoplasms: a systematic review”.

 Literature search: other databases should be explored to find eventual further articles, for example Ovid, Embase, Google scholar….

  According to your comment, “Web of Science” was added in addition to PubMed, Cochrane Library databases, and the Igaku–Chuo–Zasshi database in Japan.

 -   Please provide a reproducible search strategy with a comprehensive search string. Also, the number of key words seems to be quite low for a meaningful search

According to your comment, “large intestine” and “long period” of key words were added. Therefore, the search words were: “colorectal or colonic or large intestine” AND “endoscopic submucosal dissection or ESD” AND “long-term or long period or five-year or 5-year”. With addition to “Web of Science”, 330 citations increased to 626 citations. Finally, we included 14 studies in this review. The number of studies included papers remained unchanged at 14. The revised Figure 3 was modified.

 Why only cohort studies were included? There were no RCTs nor other prospective studies?

  The inclusion criteria of this systematic review included that the outcome was survival or recurrence rate in long-term observation. The exclusion criteria included median follow-up period of <36 months. So, 34 studies were excluded due to median follow-up period of <36 months. There were no RCTs nor other prospective studies.

Explain how were treated perforations in the retrieved articles. Also, try to detail the outcomes of those patients who experienced major complications

In the study by Cong et al., there was a total of 4 perforations (2.3%) in the 156 patients. Three of these perforations occurred during the procedure, and the patients recovered after being treated with endoscopic clipping and antibiotics. The other perforation with mild symptoms occurred 24 hours after ESD, and the patient recovered after a long period of fasting and antibiotic treatment instead of surgical intervention. Six cases of delayed bleeding (3.4%) were observed, including 1 in the ascending colon and 5 in the rectum. Bleeding in 1 of the 6 patients was stopped after conservative treatment, whereas the other patients underwent endoscopic clipping and achieved successful hemostasis. The points of bleeding were observed from the resection bed with vessels on the surface. No blood transfusions were required.

In the study by Kuwai et al., there was no perforations during the procedure in the 228 patients. One patient (0.4%) had a delayed perforation, with being treated conservatively. Delayed bleeding was observed in six cases (2.4 %) and was controlled by endoscopic hemostasis.

In the study by Takahashi et al., intraoperative perforation was observed in 20 cases (5.9%) and one case (0.3%) required emergency surgery in the non-elderly group. Intraoperative perforation and delayed perforation were observed in 10 cases (6.1%) and one case (0.6%), and the delayed perforation case required emergency surgery in the elderly group. The emergency surgery due to intraoperative bleeding was required in one case (0.3%) in the non-elderly group. The emergency surgery due to delayed bleeding was required in one case (0.6%) in the elderly group.

In the study by Boda et al., delayed bleeding after ESD occurred in 46 patients (3.7%). Intraoperative perforations occurred in 43 patients (3.4%), and 6 of these patients (0.5%) required surgery. Delayed perforations occurred in 5 patients (0.4%), and 4 of these patients (0.3%) required surgery.

In the study by Ohata et al., 7 cases (0.4%) required emergent surgery to manage delayed perforation

These points were added to the revised manuscript.

-   Add some technical details of these procedures, for example submucosal injection solution, knife used and so on….

According to your comment, we added the technical details of these procedures such as submucosal injection solution, and knife used.

  Cong et al. used a needle-knife (KD-10Q; Olympus, Tokyo, Japan), a hook-knife, and/or an insulated-tip knife. The solution for submucosal injection was 100 mL of 0.9% saline solution containing 0.4% indigo carmine and 0.0001% epinephrine.

Kuwai et al. used scissor type knives, such as the Stag-beetle Knife Jr. (SB Knife Jr.; Sumitomo Bakelite, Tokyo, Japan). The solution for submucosal injection was 0.4% sodium hyaluronate (Muco Up; Johnson & Johnson, New Jersey, USA).

Takahashi et al. used a dual knife and hook knife (Olympus Medical Systems Corp, Tokyo, Japan), and/or flush knife (Fujinon Co., Tokyo, Japan). The solution for submucosal injection was a mixture of 10% glycerol and epinephrine. When long-lasting elevation of the submucosa was required, sodium hyaluronate was injected.

  In the study by Boda et al. used a dual knife, hook knife, IT knife and flex knife (Olympus Medical Systems Corp, Tokyo, Japan), flush knife, and/or SB knife Jr. ESD was performed using 1 or 2 ESD knives. The solution for submucosal injection was 10% glycerin solution and/or .4% sodium hyaluronate.

  Ohata et al. used a dual knife and IT knife nano (Olympus Medical Systems Corp, Tokyo, Japan), flush knife, bipolar needle knife (Xeon Medical, Tokyo, Japan), and Mucosectom (Hoya. Tokyo, Japan). ESD was performed using 1 or 2 ESD knives. Sodium hyaluronate was mainly used for submucosal injection.

These points were added to the revised manuscript.

-    In Table 1, provide a pooled overall rate of the outcomes reported

According to your comment, we added the pooled overall rates of the outcomes reported. Pooled event rates (95% CIs) were 94.6% (90.5-97%) for en bloc resection, 87.1% (82.1-90.8%) for curative resection, 2.4% (1.1-5.1%) for perforation, 1.6% (0.6-4.4%) for local recurrence, 93.0% (89.6-95.4%) for five-year overall survival, and 99.4% (99.1-99.7%) for five-year overall disease-specific survival.

-     The authors should comment deeply also on the clinical outcomes of non-curative ESD, detailing the recent evidence on that topic

  We insert the factors of non-curative resection associated with long-term outcomes into Table 4. We modified and enriched the discussion on the clinical outcomes of non-curative ESD.

-      The authors should add also a paragraph on non-Asian experience with this technique

   We added a paragraph on non-Asian experience with this technique, as below.

  A recent meta-analysis showed that the outcomes of colorectal ESD in Asian countries were significantly better than those in non-Asian countries: 85.6% versus 71.3% in complete resection rate, 93% versus 81.2% in en bloc resection rate, 4.5% versus 8.6% in perforation rate, and 0.8% versus 3.1% in emergent surgery due to ESD-related adverse events [30]. This meta-analysis concluded that the ESD procedure in non-Asian countries has not yet reached adequate performance levels. In Europe, major obstacles for the dissemination of colorectal ESD include few suitable starting cases in the stomach, few experts, long learning curve, high risk of complications, and lack of standardized step-up training programs [31]. Recently, better en bloc rates of 87.9-89.5% have been reported from several groups in Europe, especially high-volume centers [32-33].  

  1. Thorlacius H, Ronnow CF, Toth E: European experience of colorectal endoscopic submucosal dissection: a systematic review of clinical efficacy and safety. Acta Oncol 2019, 58(sup1):S10-S14.
  2. Fleischmann C, Probst A, Ebigbo A, Faiss S, Schumacher B, Allgaier HP, Dumoulin FL, Steinbrueck I, Anzinger M, Marienhagen J et al: Endoscopic Submucosal Dissection in Europe: Results of 1000 Neoplastic Lesions From the German Endoscopic Submucosal Dissection Registry. Gastroenterology 2021, 161(4):1168-1178.
  3. Spadaccini M, Bourke MJ, Maselli R, Pioche M, Bhandari P, Jacques J, Haji A, Yang D, Albeniz E, Kaminski MF et al: Clinical outcome of non-curative endoscopic submucosal dissection for early colorectal cancer. Gut 2022.

 -          Figures are lacking, maybe some figures depicting the technical aspects of these procedures would be useful

  Two figures were added, and showed useful techniques such as pocket creation method and clip-with-loops method. Figure 1 shows colorectal ESD using pocket creation method. A submucosal pocket was created by dissecting the submucosa, and submucosal tunnel was created from the anal to oral side. This technique allows stable scope position and sufficient tissue traction inside the pocket. Figure 2 shows clip-with-loops method. The feature of this method is that counter traction is generated using loops, one side of which is attached to the edge of the lesion with a clip, and the other side is attached to the opposite side of the colonic wall method. Multi-loops allow to change the direction of traction by adding clip or cutting off the loop.  

Reviewer 2 Report

Reviewer's comment to the author

Thank you for giving me the opportunity of reviewing this article on Review article for long-term outcomes of colorectal endoscopic submucosal dissection.

In this study, the authors summarized several subjects related to the long-term outcomes of colorectal ESD that are clinically important. There are several points that should be discussed. I hope the revision will make this study better.

Major comments:

1.     I think that this study is mainly a list of citations, with very little discussion by the author, and lacks new knowledge that is obtained from the summarization of included articles. Please shorten the literature citation in the manuscript and summarize systemically, and more discussion for each subject (Page 3 to 8) is also needed to make a better review article. The following comments 2 to 5 are also related to this comment.

2.     Page 3, Section 3.

The authors mentioned long-term survival after colorectal ESD. The rate of cancer in the study of Cong. et al is lower than the other studies, and each study has different rates of histology. Therefore, factors that may affect survival, such as the rate of cancer, depth of disease, and vascular invasion in each study, should be added to Table 1.

3.     Page 5, Section 4.

The authors concluded that local recurrence is associated with incomplete resection and invasion into the submucosa. If there were statistically significant differences between them in studies other than Yamada's, please describe them in the manuscript and Table 2. Then, is submucosal cancer really associated with local recurrence? Not deep submucosal invasion cancer?

4.     Page 6, Section 5.

Please insert the factors of non-curative resection associated with long-term outcomes into Table 4.

5.     The authors concluded that endoscopists must improve their skills to reduce local recurrence. You may need to describe the method in a little more detail. For example, there are some tools or devices to make better for ESD procedures, such as traction and resection device. Please describe a little bit more about what methods are available at this time.

Author Response

Thank you for giving me the opportunity of reviewing this article on Review article for long-term outcomes of colorectal endoscopic submucosal dissection.

In this study, the authors summarized several subjects related to the long-term outcomes of colorectal ESD that are clinically important. There are several points that should be discussed. I hope the revision will make this study better.

 Thank you for your insightful comments, which were extremely helpful for improving the quality of our manuscript.

Major comments:

  1. I think that this study is mainly a list of citations, with very little discussion by the author, and lacks new knowledge that is obtained from the summarization of included articles. Please shorten the literature citation in the manuscript and summarize systemically, and more discussion for each subject (Page 3 to 8) is also needed to make a better review article. The following comments 2 to 5 are also related to this comment.

 According to your comments, we shortened the literature citation, and enriched the discussion. Especially, we added the pooled overall rates of outcomes in Table 1. Pooled event rates (95% CIs) were 94.6% (90.5-97%) for en bloc resection, 87.1% (82.1-90.8%) for curative resection, 2.4% (1.1-5.1%) for perforation, 1.6% (0.6-4.4%) for local recurrence, 93.0% (89.6-95.4%) for five-year overall survival, and 99.4% (99.1-99.7%) for five-year overall disease-specific survival.

  1. Page 3, Section 3.

The authors mentioned long-term survival after colorectal ESD. The rate of cancer in the study of Cong. et al is lower than the other studies, and each study has different rates of histology. Therefore, factors that may affect survival, such as the rate of cancer, depth of disease, and vascular invasion in each study, should be added to Table 1.

According to your comment, we added the factors into the Table 1.

  1. Page 5, Section 4.

The authors concluded that local recurrence is associated with incomplete resection and invasion into the submucosa. If there were statistically significant differences between them in studies other than Yamada’s, please describe them in the manuscript and Table 2. Then, is submucosal cancer really associated with local recurrence? Not deep submucosal invasion cancer?

 As you pointed out, Yamada et al. reported that deep submucosal invasion cancer associated with local recurrence. This point was modified in the revised manuscript (Section 4, the last paragraph).

  1. Page 6, Section 5.

Please insert the factors of non-curative resection associated with long-term outcomes into Table 4.

According to your comment, we added the factors into the Table 4.

  1. The authors concluded that endoscopists must improve their skills to reduce local recurrence. You may need to describe the method in a little more detail. For example, there are some tools or devices to make better for ESD procedures, such as traction and resection device. Please describe a little bit more about what methods are available at this time.

 We added two figures, and showed useful techniques such as pocket creation method and clip-with-loops method. Figure 1 shows colorectal ESD using pocket creation method. A submucosal pocket was created by dissecting the submucosa, and submucosal tunnel was created from the anal to oral side. This technique allows stable scope position and sufficient tissue traction inside the pocket. Figure 2 shows clip-with-loops method. The feature of this method is that counter traction is generated using loops, one side of which is attached to the edge of the lesion with a clip, and the other side is attached to the opposite side of the colonic wall method. Multi-loops allow to change the direction of traction by adding clip or cutting off the loop. Thank you very much.

Round 2

Reviewer 2 Report

This manuscript has been revised well. I think it will be acceptable.